# In Contrast to Anti-CCP, MMP-Degraded and Citrullinated Vimentin (VICM) Is Both a Diagnostic and a Treatment Response Biomarker

**DOI:** 10.3390/ijms24010321

**Published:** 2022-12-24

**Authors:** Patryk J. Drobinski, Neel I. Nissen, Dovile Sinkeviciute, Nicholas Willumsen, Morten A. Karsdal, Anne C. Bay-Jensen

**Affiliations:** 1ImmunoScience, Nordic Bioscience, Herlev Hovedgade 207, 2730 Herlev, Denmark; 2Department of Biomedical Sciences, University of Copenhagen, Blegdamsvej 3, 2200 Copenhagen, Denmark; 3Oncology, Nordic Bioscience, Herlev Hovedgade 207, 2730 Herlev, Denmark

**Keywords:** rheumatoid arthritis, autoimmunity, vimentin, biomarkers, low disease activity, pharmacodynamics, anti-CCP

## Abstract

Protein citrullination and degradation by matrix metalloproteinases (MMP) plays a central role in the pathology of rheumatoid arthritis (RA). Autoantibodies are known to target citrullinated vimentin. The aim of this study was to investigate the relationship between the blood levels of MMP-degraded and citrullinated vimentin (VICM), as compared with the levels of MMP-degraded and non-citrullinated vimentin (VIM), and the standard anti-CCP biomarker in RA patients undergoing treatment. Thus, VIM, VICM and anti-CCP were quantified by ELISA in serum samples from baseline and week 8 of patients (n = 257) with RA, treated with either tocilizumab (8 mg/kg), methotrexate (7.5–15 mg/kg) or a placebo and compared with a reference cohort (n = 64). The three biomarkers were elevated in RA serum compared with the reference cohort: medians were 1.7 vs. 0.8 ng/mL (*p* < 0.05) for VIM; 7.5 vs. 0.7 ng/mL (*p* < 0.0001) for VICM; 57 vs. 4 RU/mL (*p* < 0.001) for anti-CCP. VICM was decreased in response to tocilizumab (2.9-fold, *p* < 0.0001) and to methotrexate (1.5-fold, *p* < 0.05) compared with the placebo, while anti-CCP was not. Serum VIM was also modulated by both drugs, although to a lesser degree. A high baseline level of VICM was predictive of a low disease activity response at week 8. In conclusion, VICM can differentiate between RA and healthy donors in a similar manner to anti-CCP; furthermore, VICM is also a pharmacodynamic marker.

## 1. Introduction

Rheumatoid arthritis (RA) is an autoimmune disease characterized by systemic inflammation, which leads to swollen, tender joints, and bone nodulations and erosions [1]. RA is highly heterogeneous, meaning patients display a variety of molecular and phenotypic manifestations, such as different rates of disease development, progression, levels of pain and response to treatment [2,3,4]. Protein citrullination is the most specific post-translational modification (PTM) and a known trigger of autoimmunity in RA. Anti-citrullinated protein antibodies (ACPA), namely anti-CCP (cyclic citrullinated peptide) are prevalent in 70% of patients with established RA [5]. Furthermore, increased titers of circulating ACPA have been shown to be present 2–4 years prior to the onset of clinical RA [6]. The presence of autoantibodies, particularly of ACPA and rheumatoid factors (RFs), in patient circulation is a major disease hallmark and one of the main criteria for diagnosis and clinical classification of RA [7,8]. Both ACPA and RF are umbrella terms that cover different antigen species, which may arise from different pathological events. Quantification of such species may aid in profiling the molecular disease drivers that underlie the different patient subgroups. One such antigen species is citrullinated and metalloproteinase (MMP)-degraded vimentin [9].

Vimentin (NCBI ref. Seq: NP_003371.2) is a cytosolic type III intermediate filament protein involved in cell signaling, integrity and movement facilitation [10]. Vimentin can be secreted into the extracellular space by activated macrophages in RA [11,12]. Additionally, activated macrophages have demonstrated the potential to express increased levels of peptidylarginine deaminase enzyme (PAD), which accounts for the citrullination process and conversion of arginine amino acids into citrulline amino acids—a process increased in RA [13]. A biomarker assay for VICM which specifically recognizes the C-terminus of MMP-cleavage site of citrullinated vimentin has been previously established [9]. The VICM epitope is a double PTM; (i) it is citrullinated by PAD from macrophages (and, potentially, other immune cells), and (ii) it is generated via cleavage by MMPs [9]. Serum VICM has been reported to be elevated in patients with RA [14], Crohn’s disease [15], lung cancer [16] and spondyloarthritis [17]. Furthermore, serum VICM has been associated with treatment response in RA [12,18], and as a prognostic of disease progression in axial spondyloarthritis [19].

Patients included in our sub-study were a treatment-naïve cohort, subjected to two different monotherapies—methotrexate (MTX) and tocilizumab. These treatments display different modes of action and therefore different potential impacts on the modulation of inflammatory and autoimmunity manifestations. MTX, as a broad spectrum systemic immune system suppressant, acts on a wide range of immune cells and cytokines’ production [20]. This approach however often results in an inadequate response or intolerance to treatment, thus requiring alternative treatments, e.g., biological agents [21]. Tocilizumab, as opposed to MTX, limits inflammation by the selective inhibition of IL-6 receptors and preventing the IL-6 signal transduction that otherwise stimulates B and T cells [22]. It has been demonstrated that tocilizumab provides a clinically significant response in RA with a significant inhibition of joint damage progression and a significant reduction in swollen and tender joint counts [22,23,24,25]. Previous studies have shown that serum levels of VICM are inhibited by tocilizumab treatment in RA patients with moderate to severe RA [26]; however, little information is known about the levels in biologic-naïve patients. There is a medical need for the identification of biomarker candidates, which are both prognostic and modulable in the treatment of patients with early disease, to obtain molecular insight into drugs’ mechanisms of action and to make the link to disease pathogenesis.

The aim of this study was to investigate the serum levels of VICM in biologic disease-modifying anti-rheumatic drugs (bDMARDs)-naïve patients with RA and healthy donors, and to compare VICM to the levels of the non-citrullinated MMP-derived fragments of vimentin (VIM) and anti-CCP. The purpose was to understand how VICM is associated with disease activity and treatment response, and whether baseline VICM levels or early changes are predictive of the early (8 week) response to tocilizumab.

## 2. Results

### 2.1. Patient Demographics and Baseline Characteristics

The reference cohort were younger and there were fewer females in the cohort compared with the RA study (Table 1). In addition, racial composition differed between the two cohorts. Thus, age, sex and race will be considered as confounding factors in biomarker level comparisons between the RA cohort and the reference cohort.

Patients with RA in both treatment arms had comparable age, sex, BMI and disease activity (Table 1). No statistically significant differences in patient baseline clinical characteristics were observed between the three groups.

### 2.2. Serum Concentrations of VIM, VICM and Anti-CCP in Serum from RA Patients

Serum concentrations of VIM were undetectable in 13.6% and 23.9% of the reference and the RA cohorts, respectively (see Appendix Table A1). The median (IQR) VIM level was higher in patients with RA compared with the reference subjects (1.7 (0.3–3.2) vs. 0.8 (0.4–1.4), *p* = 0.016) after adjustment for age, sex and race (Figure 1A).

Serum VICM was undetectable in 93.8% of the reference cohort, but detectable in 100% of the RA cohort (Table A1). The median (IQR) VICM level was higher in the RA cohort than the reference cohort (7.5 (3.7–15.6) vs. 0.7 (0.7–0.7), *p* < 0.0001) after adjustment for age, sex, and race (Figure 1B).

Anti-CCP levels were undetectable in 96.9% of the reference cohort subjects and in 23.9% of the RA patients (Table A1). The levels were, as expected, higher in the serum of patients with RA compared with the reference subjects (57 (4.1–164) vs. 4.1 (4.1–4.1), *p* < 0.0001) (Figure 1C).

The overlap between detectable levels of VICM and anti-CCP was 72.4%, while 27.6% had detectable levels of VICM but were negative for anti-CCP (Table A2).

### 2.3. Baseline Biomarkers Levels and Correlation to Clinical Activity Measures

In RA, VICM and anti-CCP were correlated with clinical activity measures after correction for multiple testing (Table 2). VICM was correlated with CRP (r = 0.47, *p* < 0.0001) and ESR (r= 0.24, *p* = 0.0001), while anti-CCP was correlated with CRP (r = 0.28, *p* < 0.0001).

Multicollinearity was assessed for all reported associations and the variance inflation factor ranged from 1.0 to 1.1 for each of the covariates, indicating a low level of collinearity. In addition, the goodness of the fit was assessed including all covariates as independent variables against the dependent variable. The R^2^ ranged from 0.01 to 0.04, indicating a low influence of the covariates.

### 2.4. Treatment-Dependent Biomarker Modulation at Week 8

Serum VIM levels were lower at week 8 in patients treated with tocilizumab compared with those treated either placebo (1.79-fold, *p* < 0.0001) or methotrexate (1.55-fold, *p* < 0.0001) (Figure 2A). A similar pattern was observed for serum VICM, where levels were lower in tocilizumab-treated patients compared with those treated with a placebo (2.88-fold, *p* < 0.0001) or methotrexate (1.92-fold, *p* < 0.0001) (Figure 2B). In addition, VICM levels were also lower in methotrexate-treated patients compared with those given the placebo (1.50-fold, *p* < 0.05) (Figure 2B). There was no difference in the levels of anti-CCP between the treatment arms (Figure 2C).

### 2.5. Prediction of Response (or Lack of Response)

Patients were divided into tertiles (low, moderate, high) based on their level of the biomarker at baseline and at week 8 after treatment initiation. The first tertile (low levels) represents 33% of the patients with the lowest level of the biomarker (the reference group), while the third tertile represents the 33% of the patients with the highest level of the biomarker (the high group). The middle tertile represents the patients with the median level of the biomarker (the moderate group). Response (yes/no) was defined as either 20, 50 or 70% improvement or low disease activity according to the ACR-EULAR criteria [27]. There were 42% ACR20, 21% ACR50, 7% ACR70 and 10% LDA responders at week 8 (Table 3). Data on biomarker ranges can be found in Table A3.

At baseline, only VICM was predictive of response at week 8 (Table 3). Patients in the second and third tertiles were 4.2 times more likely to achieve a low disease activity (LDA) at week 8. At week 8, patients in the third tertile of VICM were 0.6, 0.2, 0.06 and 0.07 times less likely to be an ACR20, ACR50, ACR70 or LDA responder, respectively. In addition, patients in the second tertile were 0.24 time less likely to achieve LDA (Table 3). The second and third tertiles of anti-CCP at baseline were 0.12 and 0.03 times less likely to achieve LDA (Table 3). In summary, high levels of VICM at baseline were predictive of the LDA response, while low levels (derived from the treatment effect) of VICM and anti-CCP at week 8 after treatment were predictive of the response at week 8.

## 3. Discussion

In the present study we aimed to characterize the blood marker VICM—a measure of citrullinated and MMP-degraded vimentin—in the context of RA. The patients included in our sub-study were treatment-naïve, subjected to two different monotherapies: methotrexate and tocilizumab, which differ in their modes of action and impact on immune system modulation. Our objectives were to compare VICM levels to those of its non-citrullinated (VIM) form and to anti-CCP, in order to understand the potential differences and specificity of VICM, as well as its association with the tocilizumab response.

Our results showed that VICM was highly elevated in RA compared with the reference cohort, as was anti-CCP. These findings were in accordance with previous studies, which reported the presence of elevated levels of VICM [12,28], anti-CCP and RF [29,30,31] in patients with RA. Moreover, VICM and anti-CCP were not detectable in the majority of healthy subjects, suggesting their high disease specificity. In contrast, there was an overlap in VIM levels, but not VICM, between the two cohorts, supporting the previous notion that citrullination is an important post-translational modification in RA [28]. Interestingly, while 23.9% patients with RA had undetectable levels of anti-CCP at baseline, VICM was detectable in all of the patients. Thus, a precise detection of VICM might provide an additional aid in diagnosis, especially in patients that cannot be diagnosed based on conventional serostatus markers.

Previously, it has been shown that VICM can be successfully used for monitoring therapies involving tocilizumab or methotrexate [28], and other anti-inflammatory agents such as mavrilimumab [12]. The results of our sub-study (Paragraph 2.2, Figure 2) showed that, in patients with RA, VIM and VICM displayed different levels in tocilizumab vs. methotrexate. Specifically, tocilizumab showed a higher suppression of VIM and VICM compared with methotrexate, with VICM being the most suppressed. This suggests that tocilizumab may limit the inflammatory processes more effectively than methotrexate and that VICM can be used as a tool for monitoring the drug’s mode of action. Furthermore, there was no difference in the anti-CCP levels between the treatment arms, which suggests that VICM may be superior to anti-CCP for monitoring treatment effects.

For the best outcome for patients, it is important to have a prognostic biomarker towards long-term treatment efficacy as early as possible. VICM and anti-CCP have been previously reported to show a prognostic capacity for disease progression. The results of our study showed that VICM already correlated with markers of disease activity at baseline, while anti-CCP did not. Additionally, at baseline, only VICM was predictive of the treatment response at week 8. Therefore, VICM might be an early indicator of whether a patient will respond and benefit from therapy in the future. Previous published work has shown that the level of VICM is already inhibited after 4 weeks and is stabilized until 24 weeks by treatment with tocilizumab or mavrilimumab [15,26]. Thus, our results lead to the question of whether baseline or early changes in VICM predict a sustained response over 24 weeks or even 52 weeks.

The main limitation of the study is that this is a post-hoc and explorative analysis; thus, the results are indicative and need to be verified and validated in prospective designed studies. In addition, the reference cohort was from other sample sites regarding where the RA samples were acquired from; thus, there are differences in the handling of the samples. We have tested biological technical variation for the marker VICM, and no red flags were raised that serum handling had an effect on the level of the biomarker. On the other hand, the RA samples were not from the same site, as these samples were obtained from a phase III clinical trial with multiple international recruitment centers.

## 4. Conclusions

In conclusion, VICM has diagnostic potential, as its levels were elevated in RA and barely detectable in healthy donors. Furthermore, in our sub-study, VICM performed better than anti-CCP in monitoring the treatment response and in the identification of patients likely to benefit from continuation of the treatment. Therefore, VICM may be a step further in the diagnosis of RA and the development of effective personalized treatment.

## 5. Materials and Methods

### 5.1. Study Design

RA patients who were bDMARDs-naïve (N = 673) were enrolled in a 24-week randomized, double-blind, placebo-controlled phase III clinical trial testing the efficacy of 8 mg/Kg tocilizumab monotherapy (every four weeks) vs. methotrexate (MTX) monotherapy (MTX) (weekly dose, starting at dose 7.5 mg and titrated to 15 mg at week 4) and placebo (AMBITION study; NCT00109408). A detailed description of the study was published by Jones et al. in 2008 [32]. Briefly, patients above 18 years of age with active RA and with at least 3 months since diagnosis were enrolled. Active RA was defined as: SJC66 ≥ 6, TJC68 ≥ 8 and a CRP level ≥ 1 mg/dl or ESR ≥ 28 mm/h. Additionally, patients treated with MTX in the past or who had experienced toxicity to MTX were excluded. Details of the patient characteristics can be found in Table 1. Of note, oral glucocorticoids (up to 10 mg/day prednisone or equivalent) and non-steroidal anti-inflammatory drugs were permitted if the dose was stable for ≥6 weeks. The clinical trial was completed in 2009, including 149 different recruitment sites (listed on clinicaltrial.gov) and sponsored by Hoffmann–La Roche. The current biomarker study is an explorative and retrospective sub-study including all patients (n = 257, appendix Figure A1) with available serum samples at baseline and week 8. Table 1 provides an overview of the patient characteristics. The biomarker sub-study was approved by the Internal Research Board (IRB) following an ongoing research agreement between the study sponsor and the research group. The study was conducted in adherence with the Declaration of Helsinki. All patients provided written, informed consent for the use of their serum samples for exploratory biomarker research before inclusion in the study.

Serum samples from 64 healthy donors were included in the study to provide a normal reference range. The samples were obtained from the qualified (ISO3485 and GLCP compliance) vendor Discovery Life Sciences Inc. (Kassel, Germany) in compliance with the Ethics Committee recommendations and all regulations, guidelines and best practices that meet or exceed the US and EU regulatory requirements. Twenty samples went missing due to human error before measurement of the VIM biomarker; thus, this marker was only measured in 44 samples.

### 5.2. Biomarker Mesaurements

Serum markers were measured in all available samples from baseline and week 8 (Figure A1). Measurements of serum VICM and VIM were performed using validated competitive ELISAs [33] (Nordic Bioscience, Herlev, Denmark), and anti-CCP was measured using a validated indirect ELISA (Euroimmun, Lubeck, Germany). All biomarker measurements were conducted according to the manufacturers’ instructions. Briefly, VICM [9], VIM [16] and anti-CCP were measured using antigen-coated 96-well plates, to which appropriate calibrators, quality controls and serum samples were added, followed by the addition of peroxidase-conjugated antibodies and incubation for either 20 h at 4 °C, or 60 min at 20 °C for anti-CCP. Subsequently, 3.3′, 5.5′-tetramethylbenzidine chromogen was used as a substrate and 0.18 M sulfuric acid as a stop solution. Plates were read at 450 nm with the reference set to 650 nm on an absorbance microplate reader (SpectraMax, Molecular Devices Corporation, Sunnycale, CA, USA). The calibration curves for all three assays were plotted using a 4-parameter logistic regression model.

All samples were measured in duplicate. The obtained values were approved following three acceptance criteria: (1) acceptance of the standard curve with all standard points at <10% coefficient of variance (CV%) and <15% relative error (%RE), within the assay analytical measurement range. The allowed exception from that criterion was one standard point outside the acceptance criteria; (2) acceptance of the quality controls (QCs) with <15% CV and within the target range of mean ± 20%, with the exception that one out of five QC specimens was allowed to have CV > 15%; (3) acceptance of samples within the measurement range and <15% CV. Samples with a CV > 15% were re-measured.

### 5.3. Statistics

A comparison of age, sex and race variables between patients with RA and normal reference range cohort was conducted using the Mann–Whitney or Chi-squared tests (Table 1). A comparison of clinical parameters was performed using the Kruskal–Wallis or Chi-squared tests (Table 1). Samples with biomarker values above the upper limit of quantitation (ULOQ) were diluted and re-measured, whereas those that were below the lower limit of quantitation (LLOQ) were given the value of LLOQ. Biomarker data were ln-transformed (Figure 1). The differences between biomarker levels of patients with RA and the reference range cohort were analyzed by ANCOVA adjusting for age, sex and race (Figure 1). Correlation analyses (Table 2) between the different RA groups were conducted by multivariate linear regression, and were adjusted for age, sex, BMI and disease duration (common covariates). Differences in biomarkers levels between treatment arms from baseline to 8 weeks were analyzed by ANCOVA adjusting for common covariates and baseline level of the individual biomarker (Figure 2). Logistic regression was used to calculate odds ratios (ORs) for the clinical response adjusting for common covariates and baseline biomarker levels (Table 3). Significance was considered when *p* values were * *p* < 0.05, ** *p* < 0.01, *** *p* < 0.001, **** *p* < 0.0001 after Bonferroni correction. All statistical analyses were performed with MedCalc version 19.3 and Prism GraphPad version 9.0 for graphical preparation.

## Figures and Tables

**Figure 1 ijms-24-00321-f001:**
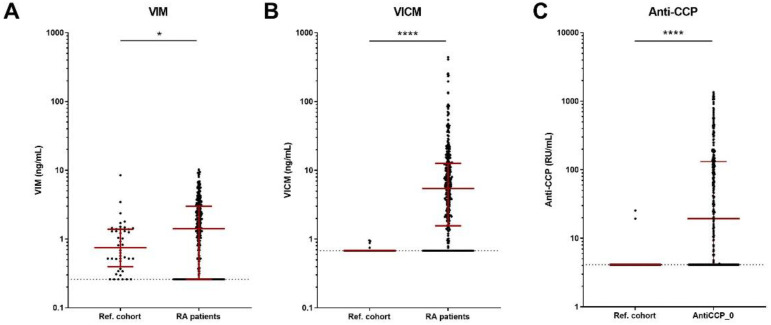
Comparing the level of markers between the reference cohort (n = 64) with the AMBITION patients (n = 257): (**A**) VIM (n_vim_ = 44 subjects (see Methods section)).; (**B**) VICM; (**C**) anti-CCP. Comparisons between groups were conducted on ln-transformed biomarker data by ANCOVA adjusting for age, sex and race. Data are depicted as scattergrams with median and IQR in red. The dotted line indicates the lower detection limit of the assay. *, *p* < 0.05 and ****, *p* < 0.0001.

**Figure 2 ijms-24-00321-f002:**
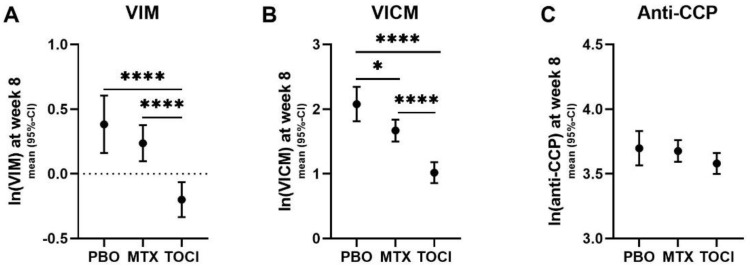
Serum levels of the biomarkers VIM, VICM and anti-CCP at week 8 post-treatment with placebo (PBO), methotrexate (MTX) or tocilizumab (TOCI). ANCOVA was used to compare between groups. Data are shown as estimated marginal means (95% CI) derived from ln-transformed data adjusted for age, sex, race, BMI, disease duration and baseline biomarker levels for the individual marker: *, *p* < 0.05 and ****, *p* < 0.0001.

**Table 1 ijms-24-00321-t001:** Patient demographics overview. Assessment of differences between the normal reference range cohort and the AMBITION study (baseline), and between the treatment arms of the AMBITION study. Data are shown as median (IQR) unless other noted. The differences were tested using non-parametric tests, either by § Mann–Whitney test, * Chi-squared test or ¤ Kruskal–Wallis test. ns, non-significant.

Variables	Ref. Cohort	RA	*p*-Value	TOCI	MTX	Placebo	*p*-Value
**N**	64	257	-	112	104	41	-
**Age (years)**	42.5 (31.0–55.0)	52.0 (42.8–59.0)	<0.0001 §	50.5 (42.5–58.5)	53.0 (42.5–61.0)	51.0 (42.3–58.3)	ns ¤
**Female, n (%)**	23 (35.9)	200 (77.8)	<0.0001 *	92 (82)	79 (76)	29 (71)	ns *
**Race** Asian Black Other White	0 21 0 43	5 10 32 210	<0.0001 *	2 4 16 90	2 5 15 82	1 1 1 38	ns *
**BMI**	-	27.9 (6.5)	-	28.5 (7.6)	27.3 (6.1)	28.5 (7.6)	ns ¤
**RADUR**	-	3.0 (0.6–10.9)	-	2.6 (0.6–8.2)	4.1 (0.7–11.7)	1.8 (0.5–15.3)	ns ¤
**CRP (mg/dL)**	<1.0	1.8 (0.8–4.0)	-	1.7 (0.7–4.3)	2.2 (0.9–4.1)	1.7 (0.6–3.5)	ns ¤
**ESR (mm/h)**	-	42.0 (32.0–58.3)	-	40.0 (30.0–56.5)	43.5 (34.5–60.5)	42.0 (31.8–56.5)	ns ¤
**DAS28 (units)**	-	6.8 (6.2–7.4)	-	6.7 (6.2–7.3)	6.8 (6.3–7.4)	6.9 (6.3–7.2)	ns ¤
**TJC (n)**	-	30.0 (20.0–41.3)	-	30.0 (19.5–41.0)	31.0 (21.0–42.0)	31.0 (24.5–45)	ns ¤
**SJC (n)**	-	16.0 (11.0–24.0)	-	16.0 (11.0–22.5)	18.0 (11.0–23.0)	17.0 (12.0–28.3)	ns ¤
**HAQ-DI (n)**	-	1.5 (1.1–2.0)	-	1.5 (1.1–2.0)	1.5 (1.9–2.0)	1.4 (1.1–1.7)	ns ¤
**Pain VAS (mm)**	-	62.0 (48.0–77.0)	-	59.5 (44.5–75.5)	65 (49.5–78.5)	61.0 (49.8–79.3)	ns ¤
**Patient global VAS (mm)**	-	68.0 (51.0–80.0)	-	67.0 (49.5–79.0)	65.0 (50.5–81.0)	70.0 (56.0–77.5)	ns ¤
**Physician global VAS (mm)**	-	67.0(54.0–77.0)	-	65.0 (52.0–76.0)	66.0 (55.0–76.8)	73.0 (58.0–81.5)	ns ¤

Abbreviations: RA, rheumatoid arthritis, TOCI, tocilizumab; MTX, methotrexate; RADUR, RA disease duration; DAS28, 28-Joint Disease Activity Score; TJC, tender joint count; SJC, swollen joint count; CRP, C-reactive protein; ESR, erythrocyte sedimentation rate; HAQ-DI, Health Assessment Questionnaire-Disability Index; VAS, Visual Analogue Scale.

**Table 2 ijms-24-00321-t002:** Multivariate linear regression between baseline biomarkers and clinical measures of disease activity. Data are shown as a partial correlation (r) adjusted for age, sex, BMI and disease duration. The serological biomarker data were log-transformed. Bonferroni-adjusted alpha < 0.0083. ns, non-significant (*p* > 0.0083).

Biomarker	DAS28	TJC	SJC	CRP	ESR	HAQ
VIM	0.05 ns	r = 0.01 ns	r = −0.07 ns	r =−0.07 ns	r = 0.10 ns	r = −0.01 ns
VICM	r = 0.13 ns	r = 0.01 ns	r = −0.01 ns	r = 0.47 *p* < 0.0001	r = 0.24 *p* = 0.0001	r = 0.14 ns
Anti-CCP	r = 0.11 ns	r = 0.03 ns	r = 0.09 ns	r = 0.28 *p* < 0.0001	r = 0.08 ns	r = 0.11 ns

DAS28, 28-Joint Disease Activity Score; TJC, tender joint count; SJC, swollen joint count; CRP, C-reactive protein; ESR, erythrocyte sedimentation rate; HAQ, Health Assessment Questionnaire-Disability Index.

**Table 3 ijms-24-00321-t003:** Baseline prediction of response using the biomarkers including all patients. OR [95% CI] for being a responder from logistic regression comparing patients with the lowest levels of biomarker (first tertile) with the second and third tertiles at either baseline or at week 8. ORs were adjusted for baseline biomarker level (week 8 only), age, sex, BMI and disease duration. Ns, non-significant (*p* > 0.05).

Number of Responders (%)	ACR20 42.1%		ACR50 21.3%		ACR70 7.1% q		LDA 10.3%	
*Baseline*	Moderate	High	Moderate	High	Moderate	High	Moderate	High
**VIM**	0.73 [0.39–1.38] ns	1.21 [0.65–2.27] ns	0.49 [0.21–1.15] ns	1.63 [0.79–3.39] ns	0.56 [0.15–2.03] ns	1.12 [0.36–3.45] ns	3.34 [0.87–12.9] ns	0.82 [0.32–2.08] ns
**VICM**	0.81 [0.43–1.52] ns	0.78 [0.42–1.46] ns	0.65 [0.30–1.40] ns	0.64 [0.30–1.37] ns	0.65 [0.21–2.01] ns	0.44 [0.12–1.56] ns	4.20 [1.39–12.7] *p* = 0.011	4.24 [1.39–13.0] *p* = 0.011
**Anti-CCP**	1.14 [0.61–2.17] ns	1.48 [0.80–2.76] ns	0.96 [0.45–2.07] ns	1.01 [0.48–2.12] ns	0.56 [0.17–1.83] ns	0.57 [0.18–1.85] ns	1.02 [0.37–2.78] ns	1.24 [0.43–3.56] ns
** *Biomarker level at Week 8* **	**Moderate**	**High**	**Moderate**	**High**	**Moderate**	**High**	**Moderate**	**High**
**VIM**	1.24 [0.59–2.61] ns	0.96 [0.38–2.42] ns	0.83 [0.34–2.04] ns	0.58 [0.19–1.75] ns	0.53 [0.13–2.19] ns	0.48 [0.08–2.73] ns	1.03 [0.31–3.45] ns	0.31 [0.06–1.58] ns
**VICM**	0.93 [0.49–1.76] ns	0.59 [0.28–1.27] *p* = 0.0023	0.47 [0.22–1.02] *p* = 0.056	0.22 [0.08–0.58] *p* = 0.0023	0.46 [0.15–1.41] ns	0.06 [0.01–0.53] *p* = 0.012	0.24 [0.08–0.74] *p* = 0.017	0.07 [0.01–0.40] *p* = 0.0026
**Anti-CCP**	1.34 [0.46–3.88] ns	1.81 [0.32–10.18] ns	0.41 [0.11–1.54] ns	0.18 [0.02–1.56] ns	0.18 [0.02–1.36] *p* = 0.097	0.08 [0.00–1.93] ns	0.12 [0.02–0.82] *p* = 0.031	0.03 [0.00–0.61] *p* = 0.023

## Data Availability

Not applicable.

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
