# Peer review of "In Contrast to Anti-CCP, MMP-Degraded and Citrullinated Vimentin (VICM) Is Both a Diagnostic and a Treatment Response Biomarker"

_ijms, 2022, doi:10.3390/ijms24010321_

Round 1
Reviewer 1 Report
Article " In contrast to anti-CCP, MMP-degraded and citrullinated vi-2 mentin (VICM) is both a diagnostic and a treatment response 3 biomarker" was reviewed. English and grammar are proper, yet there are serious issues on statistics, data presentation, conclusions and article form. some tables in my opinion are imcomprehensible. There are several issues listed below
Major :
material and methods and results are missplaced
line 196 - if samples were missing, they should not be reported in this study
treatment with sDMARDS and bDMARDS should be given with dosages (medians), treatment with glucocorticoids shoud be given with daily dosage.
Results:
is CRP given in mg/dL? or mg/L ? what is normal range? authors should adress why median of CRP isn't statistically different from control group.
are all the data normally distributed? in case of non-normal disitribution it is better to use inter quartile range instead of SD
paragraph 2.3 when presenting multivarate regression it is advised to check for colineartity and goodnes of fit.
correlations between VIMC and CRP and ESR are very weak (r below 0,5) - have the data been checked for heteroskedasticity?
in abstract "with RA patients treated with either tocilizumab (8 mg/kg), methotrexate (7.5-20 mg/kg)" but in methods its stated that : 7,5mg-15mg -line 191
what were the enrollment criteria to this study ? -this information shoud be given
in paragraph 2.4 that statistical test were used for dependent samples? wicoxon? t-test for dependent samples?
in paragraph 2.5 what statistical test were used for dependent samples?
table 3 is incomprehensible and requires change
Discussion:
"Results of our sub-study showed that" requires citation or reference to table/methods in text
8 week observation is not standart time period for reaching remission in RA (guidelines EULAR), nor dosage 7.5mg per week of methotrexate is not a standard dosage for RA treatment - authors should adress this issue
Author Response
Comments organized by structure of the manuscript
Reviewer R1 and Reviewer R2
R1: Article " In contrast to anti-CCP, MMP-degraded and citrullinated vimentin (VICM) is both a diagnostic and a treatment response biomarker" was reviewed. English and grammar are proper, yet there are serious issues on statistics, data presentation, conclusions and article form. some tables in my opinion are imcomprehensible. There are several issues listed below
R2: The aim of the manuscript was to investigate the relationship between blood levels of MMP-degraded and citrullinated vimentin, as compared to levels of MMP-degraded and non-citrullinated vimentin, and the standard anti-CCP biomarker in RA patients treated with tocilizumab. The study is very interesting, but there are some major aspects, the authors must consider before publication.
- We thank the editor and reviewers for the opportunity to improve our manuscript. We have carefully ready the comments and suggestions for revisions and to best of ability acted on the feedback. Please find the answers to the comments below.
Abstract
- R1: The abstract must be reorganized, since is a little bit confusing, due to so many values presented in it. The abstract should be very clear and concise, since it is the first impression of an article.
- Thank you for your comment. We have reformulated several sentences in the abstract to make it clearer. After some discussions, we decided to keep the values as it allows the reader to understand the magnitude of the differences observed between RA and the reference cohort, as well as before and after treatment. We hope this is acceptable.
- R1: You should mention MTX therapy also in the abstract and introduction.
- Thank you for the suggestion. We have changed the sentence in line 25 to include “both drugs” instead of the just tocilizumab.
Introduction
- R1: The introduction can be improved, since there are some missing data, such as:
- RA treatment methods and plans;
- tocilizumab therapy;
- methotrexate therapy;
- Thank you for that suggestion for improvement. We have updated the introduction with following section, which provides an introduction to the therapies included in this study:
- Patients included in our sub-study were treatment-naïve cohort, subjected to two different monotherapies Methotrexate (MTX) and Tocilizumab. These treatments dis-play different mode of action and therefore different potential impact on modulation of inflammatory and autoimmunity manifestations. MTX as a broad spectrum systemic immune system suppressant acts on a wide range of immune cells and cytokines pro-duction [21]. This approach however often results in inadequate response or intolerance to treatment, thus requiring alternative treatments e.g., biological agents [22]. Tocilizumab as opposed to MTX limits the inflammation by selective inhibition of IL-6 receptors and preventing IL-6 signal transduction that stimulates B and T cells [23]. It was demonstrated that tocilizumab provides a clinically significant responses in RA with significant inhibition of joint damage progression and significant reduction of swollen and tender joint counts [23–26].
- R1: Why have you chosen to study VIM, VICM and anti-CCP biomarkers at patients treated with tocilizumab and not other drug?- and so on.
- Previous studies have shown that VICM was associated with tocilizumab response in moderate to severe RA patients with inadequate response to DMARDs. We wanted to investigate the effect in naïve patient and VICMs ability to predict early response. We have added following sentence to line 74.
- Previous studies have shown that serum levels of VICM are inhibited by tocilizumab treatment in RA patients with moderate to severe RA [27], however little information is known about the levels in biologic naïve patients.
- It would be interesting to also test other drugs as well and this is what we plan to do. As a first approach it was sensible to test the hypothesis in a tocilizumab trial as IL-6 is known to regulate MMP activity, which is responsible for the release of VICM.
- Previous studies have shown that VICM was associated with tocilizumab response in moderate to severe RA patients with inadequate response to DMARDs. We wanted to investigate the effect in naïve patient and VICMs ability to predict early response. We have added following sentence to line 74.
- R1: Please add and emphasize the novelty/innovative aspects of this research work in the last paragraph of the introduction.
- Thank for that suggestion. Following above sentence, we have added following to line 76:
- There is a medical need for identification of biomarker candidates, which are both prognostic and modulable by treatments in patients with early disease, to get molecular insight to drug mechanism of action and make the link to disease pathogenesis.
- Thank for that suggestion. Following above sentence, we have added following to line 76:
- Line 63 – Please add the full word for bDMARDs: biologic disease-modifying antirheumatic drugs.
- Thank you. We have corrected this
- R1: You should mention MTX therapy also in the abstract and introduction.
- See point 4.
Methods
- R2: material and methods and results are misplaced
- We have used the template provided by the journal, where results are placed before method section etc.
- R2: Treatment with sDMARDS and bDMARDS should be given with dosages (medians), treatment with glucocorticoids shoud be given with daily dosage. in abstract "with RA patients treated with either tocilizumab (8 mg/kg), methotrexate (7.5-20 mg/kg)" but in methods its stated that : 7,5mg-15mg -line 191
- Thank you for noticing the typo, which has now been corrected.
- In addition, follow was added to the method section line 257:
- Of note oral glucocorticoids (up to 10 mg/day prednisone or equivalent) and non-steroidal anti-inflammatory drugs were permitted if the dose was stable for ⩾6 weeks.
- R2: Line 188 – RA patients which were bDMARDs-naive (N = 673) were enrolled in a 24-week randomized, double-blind, placebo-controlled phase III clinical trial – from where? You should mention the participating institutions.
- Following was added to the method section line 259:
- The clinical trial was completed in 2009, including 149 different recruitment sites (listed on clinicaltrial.gov) and sponsored by Hoffmann-La Roche
- Following was added to the method section line 259:
Results
- R2: is CRP given in mg/dL? or mg/L ? what is normal range? authors should address why median of CRP isn't statistically different from control group.
- CRP is measured as mg/dL. This not hsCRP, but standard CRP measurement.
- Samples from healthy controls are unmeasurable in CRP
- As the level of CRP is unmeasurable in the reference cohort then it is not possible to conduct statistical meaningful analysis
- R2: are all the data normally distributed? in case of non-normal distribution it is better to use inter quartile range instead of SD
- The biomarker data is represented as IQR in figure 1.
- Values in table 1 has been changed to median (IQR)
- R2: paragraph 2.3 when presenting multivarate regression it is advised to check for colineartity and goodnes of fit.
- Thank you for that comment. We did include it in the analyses (part of the statistical package) but did not report it. We have now included following sentence line 136:
- Multicollinearity was assessed for all reported associations and the variance inflation factor ranged from 1.0 to 1.1 for each of the covariates, indicating low level of collinearity. In addition, the goodness of the fit was assessed including all covariates as independent variables against the dependent variable. The R2 ranged from 0.01 to 0.04, indicating low influence of the covariates.
- Thank you for that comment. We did include it in the analyses (part of the statistical package) but did not report it. We have now included following sentence line 136:
- R2: correlations between VIMC and CRP and ESR are very weak (r below 0,5) - have the data been checked for heteroskedasticity?
- Thank for that suggestion. We double check the normality of the CRP and ESR and decided to include those to data as log-transformed to reduce the variance with increasing level of the markers. Only CRP and ESR were affected.
- We corrected the data in table 2 as well as the corresponding text. Also, we refined the table label to include that we transformed the biomarker data.
- Thank for that suggestion. We double check the normality of the CRP and ESR and decided to include those to data as log-transformed to reduce the variance with increasing level of the markers. Only CRP and ESR were affected.
- what were the enrollment criteria to this study ? -this information shoud be given
- The study details have been published several times. We have added following sentence to the methods, allowing reader more easily to find the original reference:
- Detailed description of the study was published by Jones et al. in 2008 [29]. Briefly, patients above 18 years of age, with active RA and with at least 3 months since diagnose were enrolled. Active RA was defined as: SJC66⩾6, TJC68⩾8 and a CRP level ⩾1 mg/dl or ESR⩾28 mm/h. Also, patients treated with MTX within the last or had experienced toxicity to MTX was excluded. Details to the patient characteristics can be found in table 1. Of note oral glucocorticoids (up to 10 mg/day prednisone or equivalent) and non-steroidal anti-inflammatory drugs were permitted if the dose was stable for ⩾6 weeks.
- The study details have been published several times. We have added following sentence to the methods, allowing reader more easily to find the original reference:
- in paragraph 2.4 that statistical test were used for dependent samples? wicoxon? t-test for dependent samples?
- We have added this sentence to the figure 2 legend:
- ANCOVA was used to compare between groups.
- We have added this sentence to the figure 2 legend:
- in paragraph 2.5 what statistical test were used for dependent samples?
- We have added following sentence to line 172 to provide more insight to what the dependent variable is:
- Response (yes/no) was defined as either 20, 50 or 70 % improvement or low disease activity according to the ACR-EULAR criteria [28].
- We have added following sentence to line 172 to provide more insight to what the dependent variable is:
- table 3 is incomprehensible and requires change
- We are not sure what the reviewer finds incomprehensible. We have updated the text in paragraph 2.5 to provide more explanation. Moreover, we updated some of the wording in the legend of and in table 3 to provide more details. We hope that this is what the reviewer was suggesting.
Discussion
- R1: 5. Line 166 – Please add the full word for MTX.
- Corrected
- R2: line 196 - if samples were missing, they should not be reported in this study
- This is a post-hoc and exploratory analysis with no prospective design. We considered that it was important to ramp up the number of samples for the two central markers (VICM and anti-CCP). We also decided not to do imputations, as this part of the study (paragraph 2.2) is descriptive.
- We did check whether biomarkers levels (VICM and anti-CCP) levels are different with and without the 20 samples. As most of the samples of reference group are measures below the detection limit for the two assays this result is not unexpected. The missing samples is only relevant for figure 1.
- R2: "Results of our sub-study showed that" requires citation or reference to table/methods in text
- Inserted
- R2: 8 week observation is not standart time period for reaching remission in RA (guidelines EULAR), nor dosage 7.5mg per week of methotrexate is not a standard dosage for RA treatment - authors should adress this issue
- This is monotherapy study, which means that the time on placebo should be restricted. At week 8, placebo cross over to active treatment. The purpose of the study was not to understand the clinical trial (prior publications describe this), rather we wanted to explore the level of VICM in biologic naïve RA patients and how the levels change in response to tocilizumab. And whether these levels are associated with early response to treatment. It would indeed be interesting to acquire an additional study where we can explore whether the levels are predictive of long term or sustained response. We have added following to the discussion:
- Previous published work show that the level of VICM is inhibited already after 4 weeks and is stabilized until 24 weeks by treatment with tocilizumab or mavrililumab [30,31]. Thus, our results leads to the question whether baseline or early changes in VICM predict sustained response over 24 weeks or even 52 weeks.
- In the aims:
- MTX is scaled from 7.5 mg to 15 mg as indicated in the method section.
- Corrected in the text
- This is monotherapy study, which means that the time on placebo should be restricted. At week 8, placebo cross over to active treatment. The purpose of the study was not to understand the clinical trial (prior publications describe this), rather we wanted to explore the level of VICM in biologic naïve RA patients and how the levels change in response to tocilizumab. And whether these levels are associated with early response to treatment. It would indeed be interesting to acquire an additional study where we can explore whether the levels are predictive of long term or sustained response. We have added following to the discussion:
- R1: Even if conclusion section is not mandatory, I suggest you to add one.
- We have added a headline call conclusion. However, if editor believe it should be removed then we can remove it again.
References
- The references must be written in the style provided.
- Corrected

Reviewer 2 Report
The aim of the manuscript was to investigate the relationship between blood levels of MMP-degraded and citrullinated vimentin, as compared to levels of MMP-degraded and non-citrullinated vimentin, and the standard anti-CCP biomarker in RA patients treated with tocilizumab. The study is very interesting, but there are some major aspects, the authors must consider before publication.
1. The abstract must be reorganized, since is a little bit confusing, due to so many values presented in it. The abstract should be very clear and concise, since it is the first impression of an article.
2. The introduction can be improved, since there are some missing data, such as:
- RA treatment methods and plans;
- tocilizumab therapy;
- methotrexate therapy;
- why have you chosen to study VIM, VICM and anti-CCP biomarkers at patients treated with tocilizumab and not other drug?
- and so on.
3. Please add and emphasize the novelty/innovative aspects of this research work in the last paragraph of the introduction.
4. Line 63 – Please add the full word for bDMARDs: biologic disease-modifying antirheumatic drugs.
5. Line 166 – Please add the full word for MTX.
6. You should mention MTX therapy also in the abstract and introduction.
7. The references must be written in the style provided.
8. Line 188 – RA patients which were bDMARDs-naive (N = 673) were enrolled in a 24-week randomized, double-blind, placebo-controlled phase III clinical trial – from where? You should mention the participating institutions.
9. Even if conclusion section is not mandatory, I suggest you to add one.
Author Response

(The authors gave the same response as above.)
